# Native Parallel Reasoner: Reasoning in Parallelism via Self-Distilled Reinforcement Learning

**Tong Wu** [* 1]  **Yang Liu** [* 1]  **Jun Bai** [* 1]  **Zixia Jia** [1]  **Shuyi Zhang** [1]  **Ziyong Lin** [1]  **Yanting Wang** [1]
**Song-Chun Zhu** [1]  **Zilong Zheng** [1]

## Abstract

We introduce **Native Parallel Reasoner (NPR)**, a teacher-free framework that enables Large Language Models (LLMs) to self-evolve genuine parallel reasoning capabilities. NPR transforms the model from sequential emulation to native parallel cognition through three key innovations: (1) a **self-distilled** progressive training paradigm that transitions from cold-start format discovery to strict topological constraints without external supervision; (2) a novel **Parallel-Aware Policy Optimization (PAPO)** algorithm that optimizes branching policies directly within the execution graph, allowing the model to learn adaptive decomposition via trial and error; and (3) a robust **NPR Engine** that refactors memory management and flow control of SGLang to enable stable, large-scale parallel RL training. Across eight reasoning benchmarks, NPR trained on Qwen3-4B achieves performance gains of up to 24.5% and inference speedups up to 4.6×. Unlike prior baselines that often fall back to autoregressive decoding, NPR demonstrates 100% genuine parallel execution, establishing a new standard for self-evolving, efficient, and scalable agentic reasoning.

## 1. Introduction

The advent of super-scale Large Language Models (LLMs), exemplified by Gemini 3 (Pichai et al., 2025), GPT-5 (OpenAI, 2025), and DeepSeek-V3.2 (DeepSeek, Inc., 2025), has shifted the frontier of AI from semantic fluency to deep, multi-step agentic reasoning. Despite the excitement of *"deeper"* test-time scaling that enables models to solve complex problems (Muennighoff et al., 2025), the *"wider"* reasoning capacity to explore diverse trajectories in parallel emerges as the dominant requirement toward agentic AI (Shen et al., 2025; Comanici et al., 2025). The MapReduce paradigm has long underpinned distributed computing by separating task decomposition from trajectory aggregation (Dean & Ghemawat, 2008; Yang et al., 2025b; Wang et al., 2025a), yet its application to agentic language modeling remains a critical missing link in the evolution of open-source LLMs. Ideally, the model should internalize the collaborative breadth of multi-agent systems directly into an efficient, natively parallel architecture.

Despite this imperative, existing implementations remain fragmented, hindering parallel reasoning as an intrinsic model capability and leading to three critical limitations. First, **Algorithmic and Architectural Incompatibility.** Existing inference engines (Zheng et al., 2024; Kwon et al., 2023) and Reinforcement Learning (RL) algorithms (Shao et al., 2024) are not designed for native branching. Inference engines lack fine-grained control over parallel branch creation and aggregation, while RL algorithms often mask or clip the gradients of the special tokens that trigger these operations, preventing models from learning strict structural patterns. Second, **Inefficient Hand-Crafted Parallelism.** Although the intuitive advantage of parallel sampling lies in efficiency, early attempts on internalizing the parallelism (Zheng et al., 2025b; Wen et al., 2025; Zhao et al., 2025) resort to hand-crafted divide-and-conquer rules via independent sampling. These methods fail to leverage shared Key-Value (KV) states, necessitating redundant recalculations for every branch and resulting in prohibitive linear latency costs that render the model impractical for real-time deployment. Third, **Reliance on Supervised Distillation.** Framework such as Multiverse (Yang et al., 2025b) successfully operationalizes native parallelism but depends heavily on supervised data distilled from stronger teacher models. While effective for compressing capabilities into smaller models, this dependence restricts the student to mimicking the teacher's sequential reasoning topology force-fitted into a parallel format, imposing an *"Intelligence Ceiling"* that prevents it from novel, model-intrinsic parallel strategies necessary for super-intelligence.

---

[*]Equal contribution  [1]State Key Laboratory of General Artificial Intelligence, BIGAI. Correspondence to: Zilong Zheng <zlzheng@bigai.ai>.

*Proceedings of the 43rd International Conference on Machine Learning*, Seoul, South Korea. PMLR 306, 2026. Copyright 2026 by the author(s).

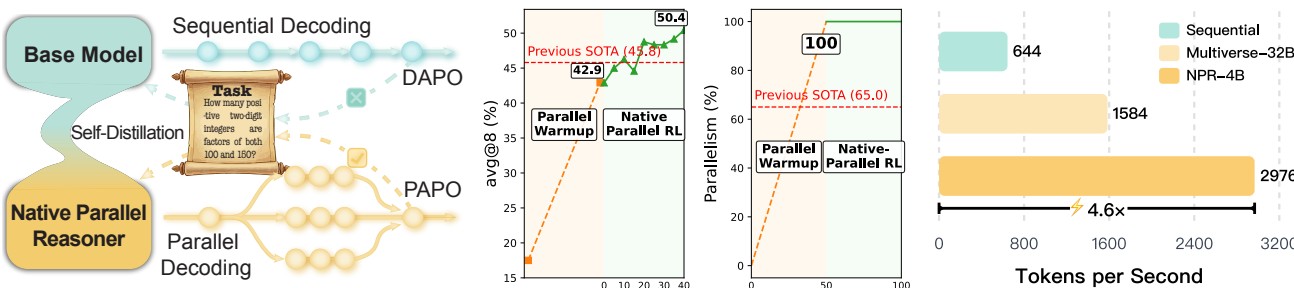

*Figure 1.* Native Parallel Reasoner (NPR) transforms a base model from sequential chain-of-thought (CoT) to native parallel reasoning via a self-distilled progressive training paradigm. Compared with previous SoTA, NPR achieves high reasoning accuracy, genuine parallelism and token acceleration. The illustrated results are evaluated on the AIME25 benchmark.

To address these challenges, we take the first step to explore the potential of LLMs to self-evolve parallel reasoning capabilities without reliance on external supervision and introduce **Native Parallel Reasoner (NPR)** (*cf.* Figure 1). Specifically, NPR employs a three-stage progressive training paradigm designed to transition the model from sequential emulation to genuine parallel cognition. In **Stage 1** (§3.2), we warm up a seed instruction-tuned model (*e.g.*, Qwen3-4B-Instruct) to spontaneously discover valid parallel structures by applying standard DAPO (Yu et al., 2025) with a format-aware reward function, yielding a structured trajectory generator, NPR-ZERO, which produces parallel-formatted outputs but still relies on sequential visibility, *i.e.*, simulated parallelism. In **Stage 2** (§3.3), we bridge the gap to native parallel architecture by performing rejection sampling on NPR-ZERO and conducting parallel warmup, which instills strict topological constraints using parallel positional encoding and attention mask as in Yang et al. (2025b), converting the sequential behavior of models into real parallel execution. Finally, **Stage 3** (§3.4) generalizes these capabilities beyond the initial distribution via Native Parallel RL. This stage implements collision-free parallel rollouts using a novel Parallel-Aware Policy Optimization (PAPO) algorithm, which optimizes branching policies directly within the parallel execution graph, allowing the model to learn adaptive decomposition strategies through trial and error rather than imitation.

To support this algorithmic breakthrough, we re-engineered the rollout infrastructure with a robust **NPR Engine** (§3.5). Specifically, we observed several stability issues unique to parallel RL, *e.g.*, GPU memory leaks caused by radix-cache mechanism; excessively long generation due to the incorrect parallel token calculations; potential runtime failure because of adaptive parallel inference logic, *etc.* **NPR Engine** re-designs memory management and flow control, providing the **first** stable rollout backend capable of supporting large-scale parallel RL.

We experiment NPR on Qwen3-4B-Instruct-2507 and Qwen3-4B (Non-Thinking) across a broad suite of eight reasoning benchmarks, demonstrating consistent

performance improvements on both original model and previously RL-tuned version up to **24.5%** and inference speedup up to **4.6×**. The results reveal three consistent superiorities of NPR (§4.3).

- **Self-Distilled Data Efficacy:** Our self-distilled datasets outperform previous teacher-generated trajectories in Yang et al. (2025b) by an average of **10.1** points, validating the hypothesis to learn from native distributions.
- **Parallelism Effectiveness and Efficiency:** Both NPR-BETA and NPR-RL yields significant performance gains over direct sequential RL baselines (*e.g.*, DAPO), confirming that adaptive parallel policies provide a superior search mechanism compared to single-path rollouts.
- **100% Genuine Parallelism:** We observed 30%+ AR fallback on test cases when running previous baselines (§5.2), where models choose run vanilla AR generation to reach better performance. In contrast, NPR performs 100% **genuinely** parallel reasoning, with no instances of hidden AR fallbacks or pseudo-parallel behavior in all evaluated test cases.

Finally, we conduct comprehensive analyses of NPR spanning inference acceleration (§5.1), pseudo-parallelism (§5.2), evolution dynamics (§5.3), test-time scalability (§B), and qualitative case studies (§C). NPR achieves task-dependent speedups, reaching up to **4.6×** over autoregressive (AR) decoding. Using *best@8* to assess test-time scalability, we find that parallel SFT and parallel RL consistently improve best-case performance across most benchmarks. Qualitative studies further show that NPR adapts its parallelism to different problem types, enabling faster and more reliable inference.

## 2. Related Work

**Parallel Reasoning.** Parallel reasoning improves reasoning efficiency and robustness by exploring multiple reasoning paths simultaneously, unlike standard sequential reasoning which is prone to early commitment errors (the "prefix

trap") and lacks self-correction, leading to suboptimal solutions and slow inference due to its strictly step-by-step generation process (Wang et al., 2025b). Early methods, such as Best-of-N (Cobbe et al., 2021) and Self-Consistency (Wang et al., 2023), select the most scored or consistent output from independent paths but are not end-to-end optimized. Search-based approaches like Tree-of-Thought (Yao et al., 2023), Graph-of-Thought (Besta et al., 2024), Monte Carlo Tree Search (Xie et al., 2024) further explore reasoning trees but rely on hand-designed structures and external verifiers, limiting flexibility and scalability. To further improve the adaptability and flexibility of parallel reasoning operations, recent work strives through learning approaches. One line of work adopts the SFT paradigm, for example, Multiverse (Yang et al., 2025b), ParaThinker (Wen et al., 2025), and SSFT (Jia et al., 2025), which guide model learning through parallel reasoning paths derived from the sequential trajectories of more powerful large reasoning models (LRMs). However, such pure imitation limits the model's ability to discover novel reasoning patterns. Another line of work enhances parallel reasoning capabilities through reinforcement learning (RL), such as APR (Pan et al., 2025) and Parallel-R1 (Zheng et al., 2025b). However, these methods either demonstrate effectiveness only on toy tasks or still depend on supervised data from other reasoning models to bootstrap the RL process.

**RL for Reasoning.** Reinforcement learning (RL) has become an important tool for enhancing the reasoning capabilities of large language models (LLMs) in recent years (Havrilla et al., 2024; Zhang et al., 2025a; Wu et al., 2025; Li et al., 2025; Liu et al., 2025a; Bai et al., 2025; Wu et al., 2024). Early and widely adopted approaches, such as RL from human feedback (RLHF), optimize outcome-level rewards derived from human preferences or task-level correctness (Meng et al., 2024). These methods improve alignment and robustness in general generation tasks but provide only coarse control over intermediate reasoning trajectories. Then, research shifts toward process-aware RL, where step-level reward modeling offer denser and more interpretable supervision (Lightman et al., 2024; Zhang et al., 2025b; Khalifa et al., 2025). Those process-level feedback, however, suffer from subjectivity, high annotation cost, and unstable optimization due to ambiguous or unverifiable intermediate signals. A further evolution leads to Reinforcement Learning with Verifiable Reward (RLVR), which replaces opaque reward models with explicit, auditable verifiers (*e.g.*, logical checkers, rule-based graders, or formal validators) (Shao et al., 2024; Xie et al., 2025; Yu et al., 2025; Zheng et al., 2025a). Compared with conventional reward modeling, RLVR provides objectivity, reproducibility, and stronger correctness guarantees, making it particularly suited for reasoning tasks where outputs are verifiable (*e.g.*, math, programming, or factual QA). More-

over, RLVR reduces human labeling costs by leveraging deterministic verifiers as reward oracles.

## 3. Native Parallel Reasoner

In this study, we propose **Native Parallel Reasoning (NPR)**, a framework that enables language models to generate and evaluate multiple reasoning branches in parallel. As shown in Figure 2, NPR is developed through a three-stage curriculum that progressively induces, grounds, and amplifies this capability. First, **NPR-ZERO** uses reinforcement learning to induce a structured parallel format without relying on external annotations. Next, **NPR-BETA** stabilizes these emerging parallel primitives through supervised fine-tuning on self-distilled trajectories. Finally, **NPR** applies a parallel-aware reinforcement learning procedure that directly optimizes the model's ability to perform native parallel reasoning. Together, these stages establish a cohesive path from initial format induction to fully optimized parallel inference.

### 3.1. Preliminaries

**Parallel Reasoning.** Parallel Reasoning (PR) relaxes the strict left-to-right dependency of AR reasoning, allowing the model to generate multiple reasoning steps independently whenever possible. Formally, the joint probability of a reasoning sample $\hat{y}$ consisting of $T$ reasoning steps $\{s_t\}_{t=1}^T$ can be factorized according to a dependency graph $\mathcal{G}$ defined over the steps:

$$P(\hat{y} \mid q; \theta) = \prod_{t=1}^T P(s_t \mid \mathrm{Pa}(s_t), q; \theta),$$

where $\mathrm{Pa}(s_t)$ denotes the set of parent steps that $s_t$ directly depends on in $\mathcal{G}$, and $\theta$ are the model parameters. This formulation enables the model to process reasoning steps that do not have mutual dependencies concurrently.

**Policy Optimization for Language Models.** To optimize the policy model within our reinforcement learning framework, we adopt objective functions based on DAPO (Yu et al., 2025). We first introduce the original DAPO update procedure. For each question-answer pair $(q, y) \sim \mathcal{D}$, the policy model $\pi_{\theta_{\mathrm{old}}}$ first generates a group of responses $\{\hat{y}_i\}_{i=1}^G$. The objective function $\mathcal{J}(\theta)$ is then formulated as:

$$\mathcal{J}(\theta) = \mathbb{E}_{(q,y)\sim\mathcal{D}, \{\hat{y}_i\}_{i=1}^G \sim \pi_{\theta_{\mathrm{old}}}(\cdot|q)}$$

$$-\frac{1}{\sum_{i=1}^G |\hat{y}_i|} \sum_{i=1}^G \sum_{t=1}^{|\hat{y}_i|} \left[ \min\left( r_{i,t}(\theta) \hat{A}_{i,t}, \right. \right.$$

$$\left. \left. \mathrm{clip}(r_{i,t}(\theta), 1 - \epsilon_{\mathrm{low}}, 1 + \epsilon_{\mathrm{high}}) \hat{A}_{i,t} \right) \right].$$

$$\text{s.t.} \quad 0 < \left| \{\hat{y}_i \mid \mathrm{is\_equivalent}(y, \hat{y}_i)\} \right| < G$$

(1)

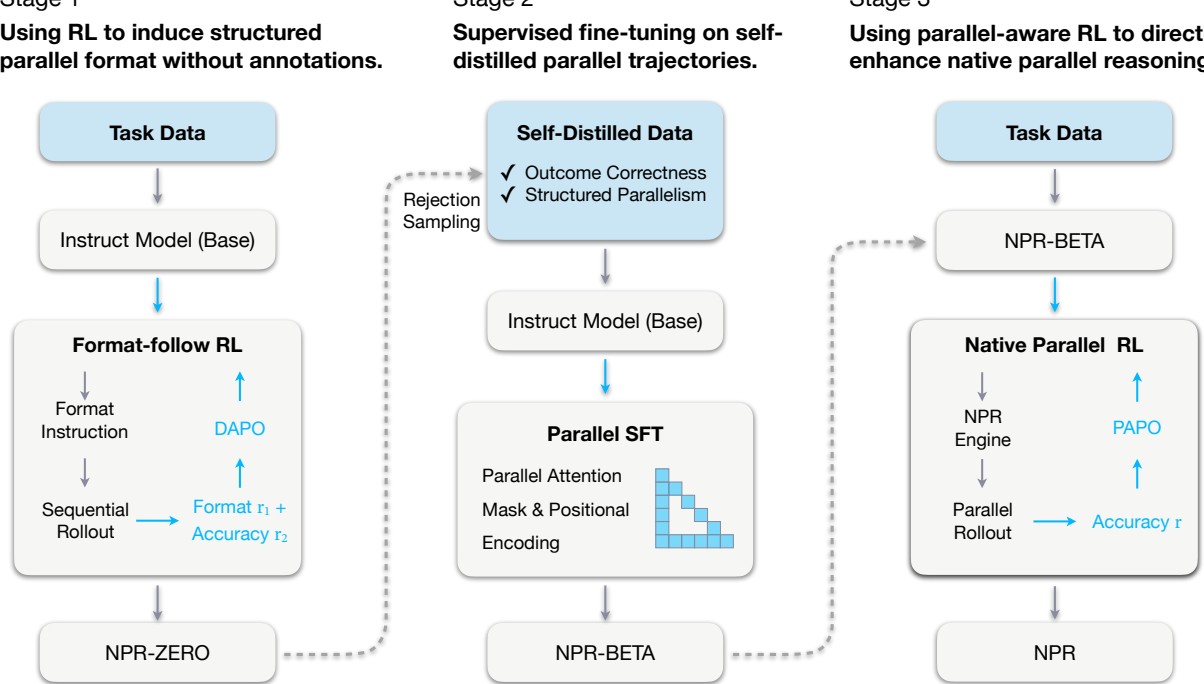

*Figure 2.* An overview of the NPR training framework.

where $r_{i,t}(\theta)$ denotes the probability ratio between the current and the old policy for the $t$-th token in response $\hat{y}_i$, and $\hat{A}_{i,t}$ represents the standardized advantage of that token computed from the rewards $\{R_1, R_2, \ldots, R_G\}$ of all generated responses in the group:

$$
\begin{aligned}
r_{i,t}(\theta) &= \frac{\pi_\theta(\hat{y}_{i,t} \mid q, \hat{y}_{i,<t})}{\pi_{\theta_{\text{old}}}(\hat{y}_{i,t} \mid q, \hat{y}_{i,<t})}, \\
\hat{A}_{i,t} &:= \frac{R_i - \text{mean}(\{R_1, R_2, \cdots, R_G\})}{\text{std}(\{R_1, R_2, \cdots, R_G\})}.
\end{aligned}
\tag{2}
$$

This formulation ensures stable policy updates by clipping extreme probability ratios while encouraging exploration through group-wise normalization of advantages. It effectively balances between exploiting high-reward responses and maintaining diversity among generated outputs.

### 3.2. Stage 1: Format-Compliant RL

To support adaptive decomposition and parallel reasoning during generation, we adopt a simplified *Map–Process–Reduce* schema, inspired by Multiverse (Yang et al., 2025b) but with a leaner structure. Each parallel block starts with `<guideline> ... </guideline>`, which contains multiple `<plan> ... </plan>` entries specifying the Map stage. The Process stage then executes each mapped subtask independently and in parallel within a separate `<step> ... </step>` block. Once all `<step>` blocks are completed, the Reduce stage consolidates their outputs into a final summary enclosed by `<takeaway> ... </takeaway>`. This explicit tag-

*Table 1.* Structured schema of NPR.

**The Output Format Example of Parallel Reasoning**

```
<guideline>
<plan>1: [One-sentence independent
strategy]</plan>
<plan>2: [One-sentence independent
strategy]</plan>
...
</guideline>
<step>1: [Self-contained detailed analysis
for plan 1]</step>
<step>2: [Self-contained detailed analysis
for plan 2]</step>
...
<takeaway>[Compare steps, synthesize
findings, determine next action]</takeaway>
<guideline>
<plan>1: [One-sentence strategy]</plan>
...
</guideline>
<step>1: [Self-contained detailed
analysis]</step>
...
<takeaway>[Final synthesis and
conclusion]</takeaway>
[Final user-facing summary. Include
\boxed{answer} for definitive short
answers.]
```

based format makes decomposition, parallel execution, and final aggregation easy to parse, verify, and supervise during downstream training and evaluation.

While this schema provides a clear, learnable format for parallel reasoning, obtaining large-scale, high-quality training data for it remains challenging. Prior work such as Multiverse (Yang et al., 2025b) constructs large, multi-step synthetic pipelines and aggregates outputs from several state-of-the-art teacher models (*e.g.*, Deepseek R1 (Guo et al., 2025) and Gemini 2.5 Pro (Google, 2025)) to overcome data scarcity. Although effective, such multi-teacher pipelines introduce considerable operational complexity: they depend on access to powerful external teacher models, require careful orchestration across heterogeneous generation processes, and incur substantial computational and maintenance costs.

We adopt a simpler, self-improving approach. Starting from a single pretrained LLM, we apply DAPO (Yu et al., 2025) to induce the target native parallel-reasoning generation format **without** paired supervision or external teachers. Our reward function combines format and accuracy signals. For format: outputs that pass a format check receive a reward of 0.0; outputs that fail receive a penalty in (0.0, -2.0]. For accuracy: when the format check passes, correct answers yield +1.0 and incorrect answers yield -1.0. The checkpoint produced by this process (denoted NPR-ZERO) is therefore optimized primarily to learn the required structured format; we then use its generations for large-scale **self-distillation** to build a synthetic corpus for downstream supervised fine-tuning (SFT).

This pipeline removes the dependency on multiple external teacher models and produces a scalable, structured dataset that supports subsequent SFT stages.

### 3.3. Stage 2: Rejection Sampling and Parallel Warmup

**Structured Trajectories Collection via Rejection Sampling.** To obtain high-quality structured reasoning traces without relying on external annotations, we employ a simple self-distillation procedure. For each question $q_i \in \{q_1, q_2, \ldots, q_N\}$ in the dataset, the model generates $K$ candidate reasoning trajectories and corresponding answers $\{(r_j^i, \hat{a}_j^i)\}_{j=1}^K$ by repeated sampling. These samples form the pool from which we extract positive supervision signals.

We apply a rejection-sampling filter designed to mirror the bootstrapping setup used in NPR-ZERO. Each sampled trajectory is evaluated using two lightweight, indicator-style constraints:

- **Outcome Correctness:** Trajectories whose predicted answer $\hat{a}$ does not match the ground-truth answer $a_i$ are discarded. This rule is represented by the indicator $\mathbb{1}_{\text{correct}}(\hat{a})$.
- **Structured Parallelism:** To ensure clean supervision for parallel generation, we remove any trajectory that fails to adhere to the required structured output format (Table 1). This constraint is encoded as $\mathbb{1}_{\text{format}}(r)$.

A sample is accepted only if it satisfies both criteria:

$$\mathbb{1}_{\text{accept}}(r, \hat{a}) = \mathbb{1}_{\text{correct}}(\hat{a}) \cdot \mathbb{1}_{\text{format}}(r). \quad (3)$$

Applying this filter yields the distilled dataset

$$\begin{aligned} \mathcal{D}_{\text{accept}} = \{(q_i, r_j^i, \hat{a}_j^i) \mid i \le N, j \le K, \\ \text{s.t. } (r_j^i, \hat{a}_j^i) \sim \pi_\theta(\cdot|q_i), \mathbb{1}_{\text{accept}}(r_j^i, \hat{a}_j^i) = 1\}. \end{aligned} \quad (4)$$

These accepted trajectories serve as the training corpus for the subsequent supervised fine-tuning stage, which provides a stable initialization for the parallel RL procedure described in §3.4.

**Parallel Attention Mask & Positional Encoding.** To support structured parallel generation, we adopt the core design of Multiverse Attention (Yang et al., 2025b) when constructing both the parallel attention mask and the corresponding positional encoding (Algorithm 1 and Algorithm 2). This design enables multiple reasoning paths to coexist within a single forward pass while allowing fast adaptation from only a few examples. It also permits efficient KV-cache reuse for the shared context inside the NPR Engine (§3.5), reducing inference overhead. Furthermore, to ensure the model can emit the required structural tags, we initialize a set of special tokens that correspond to these tags and expose them during the cold-start training stage.

**Parallel Warmup.** With the parallel mask and positional scheme in place, we perform a supervised warmup step on the distilled dataset $\mathcal{D}_{\text{accept}}$. The model is trained using standard negative log-likelihood. This stage produces the NPR-BETA, which serve as a stable initialization for the subsequent parallel reinforcement learning stage.

### 3.4. Stage 3: Native Parallel RL

While Parallel-SFT teaches the model the basic primitives of native parallel reasoning, supervised imitation alone is not sufficient. SFT-distilled trajectories tend to lack structural diversity, and some reasoning modes do not generalize beyond the training distribution. To amplify and generalize these capabilities, we introduce a dedicated Native Parallel RL stage, as shown in Figure 3. Because NPR-BETA already learns consistent parallel patterns, it serves as a reliable initialization for direct RL.

Below we summarize the practical modifications we make to standard RL (Yu et al., 2025) to respect parallel semantics and stabilize training.

**(1) Parallel Rollouts with Parallel Inference Engine.** Existing inference engines (Kwon et al., 2023; Zheng et al., 2024) do not enforce strict parallel semantics, so they can produce malformed trajectories. We therefore sample rollouts using our NPR-Engine (§3.5), which guaran-

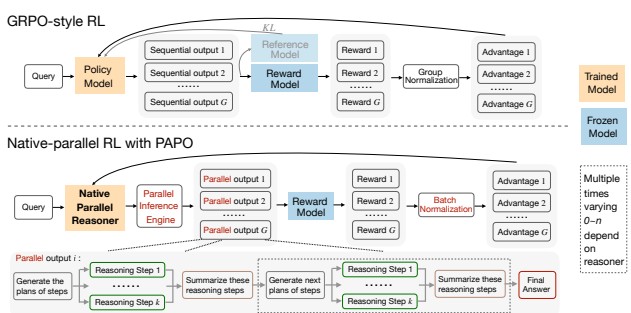

*Figure 3.* Comparison of GRPO-style RL ([Shao et al., 2024](#)) and Parallel-Aware Policy Optimization.

tees that every generated trajectory follows the intended Map–Process–Reduce flow.

**(2) Structural Filtering during Rollout.** Even with a structured engine, rare format violations can occur. To prevent malformed sequences from entering optimization, we perform **schema-level filtering** during rollout. Rather than relying solely on a text-based format checker[1], we use the SFT-constructed attention-mask and position-id encoding that exactly represent the parallel schema. After filtering, all retained rollouts strictly obey the target structure; therefore the reward reduces to accuracy only[2].

**(3) Batch-Level Advantage Normalization.** Because format-violating samples are removed before optimization, group-level variance collapses, which makes relative (group) advantages ineffective. We adopt a Lite-PPO ([Liu et al., 2025b](#)) style advantage but replace group-level variance with **batch-level** variance. For each sample $i$ and token $t$ we compute

$$\hat{A}_{i,t} := \frac{R_i - \text{mean}(\{R_1, R_2, \cdots, R_G\})}{\text{std}(\{R_1, R_2, \cdots, R_G, \cdots, R_{N*G}\})}, \quad (5)$$

where $N$ is batch size and $G$ is group size, and $R$ is the **accuracy** reward described above.

**(4) Preserve Gradients on Special Tokens.** Special tokens[3] are critical to maintain parallel semantics. Token-level clipping that suppresses gradients for these tokens breaks the learned structure, so we remove clip-masking and ensure special tokens always receive gradients. However, removing clip-masking makes importance-sampling ratios in PPO ([Schulman et al., 2017](#)) unstable. To avoid unstable reweighting, we eliminate importance sampling and adopt a strict on-policy objective. This both stabilizes training and

---

[1] We found that the text-based format checker always misses rare corner cases.

[2] +1 for a correct final answer, -1 otherwise.

[3] The tags that control parallel branching and merging.

speeds it up because we do not need to recompute historical log-probabilities.

Putting these choices together yields our Parallel-Aware Policy Optimization (PAPO) objective:

$$\mathcal{J}(\theta) = \mathbb{E}_{(q,y)\sim\mathcal{D},\,\{\hat{y}_i\}_{i=1}^{G}\sim\pi_\theta(\cdot|q)}$$
$$- \frac{1}{\sum_{i=1}^{G}|\hat{y}_i|} \sum_{i=1}^{G} \sum_{t=1}^{|\hat{y}_i|} \Big[ \frac{\pi_\theta(\hat{y}_{i,t} \mid q, \hat{y}_{i,<t})}{\text{sg}[\pi_\theta(\hat{y}_{i,t} \mid q, \hat{y}_{i,<t})]} \hat{A}_{i,t} \Big].$$
$$(6)$$

where $\text{sg}[\cdot]$ denotes stop-gradient. In practice, the stop-gradient fraction acts to preserve on-policy gradient flow while avoiding unstable importance reweighting.

### 3.5. Engineering Enhancement: NPR Engine

Multiverse's parallel-generation engine ([Yang et al., 2025b](#)) supplies a powerful substrate for large-scale rollout based on SGLang[4], but when exercised at production scale, it exposed a set of brittle implementation corners that undermine both correctness and RL stability. We implemented a compact set of engine-level mitigation to restore deterministic behavior, memory safety, and correct length accounting across high-throughput parallel rollouts, which together form the NPR-Engine used in our Parallel-RL pipeline.

**KV-Cache Double-Free and Memory Corruption.** Under heavy parallel branching, shared radix-tree KV paths were sometimes recycled more than once when the cache exceeded its capacity; the incidence scaled with the branching factor and produced context corruption and, in pathological cases, GPU memory leakage.

***Solution:*** We replace opportunistic recycling with an explicit, budget-aware reclamation strategy: when observed KV usage would exceed the preallocated budget we perform an immediate cache flush and deterministic reallocation of the affected blocks.

**Underestimated Global Token Budget.** Parallel decoding multiplies aggregate token consumption roughly by the number of branches, but the original accounting tracked only the longest single branch, allowing runs to exceed the configured `max_new_tokens`.

***Solution:*** We extended length accounting to be branch-aware: the engine now records the active branching factor at each expansion and updates a global token ledger accordingly.

**Undefined States from Illegal Parallel Schemas.** Certain parallel-branch layouts fell outside the engine's conditional logic, producing undefined states in rare corner cases.

---

[4] https://github.com/sgl-project/sglang

*Solution:* We add a lightweight pre-branch format validator that enforces a small set of structural invariants before any expansion. These checks are intentionally cheap and conservative, only structurally valid branchings are permitted, so they prevent illegal states with negligible runtime cost.

**Local Repetition inside `<step>` Blocks.** Fine-grained step streams tended to exhibit local repetition under parallel sampling, which degraded the clarity of stepwise traces.

*Solution:* We apply a mild, selective repetition penalty (coefficient = **1.02**) to tokens generated within `<step>...</step>` contexts while keeping `<guideline>` and `<takeaway>` streams penalty-neutral (1.0).

After integrating these fixes into the `verl` rollout framework, the NPR-Engine exhibited substantially improved determinism, memory stability, and correctness under large-scale parallel RL workloads. Empirical training and evaluation indicate these engine-level remedies are essential: they prevent subtle off-policy artifacts and stabilise optimization when operating at the throughput demanded by production Parallel-RL.

## 4. Experiments

### 4.1. Training Setup

**Training Datasets.** We build our experiments on the ORZ dataset (Hu et al., 2025), which contains 57k problem–answer pairs. To ensure consistency across all stages of our pipeline, we sample a fixed subset of **8k** examples from ORZ and use this for Stage 1 (§3.2), Stage 2 (§3.3), and Stage 3 (§3.4).

**Training Details.** Our models are based on Qwen3-4B-Instruct-2507 and Qwen3-4B (non-thinking mode) (Yang et al., 2025a). We intentionally avoid the thinking-mode variant because it cannot be trained with standard supervised fine-tuning.

We summarize the key configurations for each stage below.

- **Stage 1.** We follow the DAPO setup and allow a maximum generation length of 30,000 tokens.
- **Stage 2.** Training begins with a learning rate of 1e-6, which is decayed to 5e-7. We apply a weight decay of 0.1.
- **Stage 3.** We employ our PAPO together with the NPR engine. The maximum generation length remains 30,000 tokens, and the learning rate is set to 1e-7.

### 4.2. Evaluation Setup

**Evaluation Metrics.** We measure accuracy using **avg@k**, defined as the expected proportion of correct answers among

$k$ generated solutions for each problem. If the model produces $k$ candidate solutions and $c$ of them are correct, the metric reduces to

$$\text{avg@}k = \frac{c}{k}, \tag{7}$$

**Evaluation Benchmarks.** We assess the effectiveness and generalization ability of NPR across a diverse suite of reasoning benchmarks. For relatively small-scale datasets such as AIME24 (Mathematical Association of America, 2024), AIME25 (Mathematical Association of America, 2025), HMMT25 (Balunović et al., 2025), and AMC23 (Mathematical Association of America, 2023), we report **avg@8**, which better reflects performance when multiple sampled solutions are available. For larger or more heterogeneous benchmarks including OlympiadBench (He et al., 2024), Minerva-Math (Lewkowycz et al., 2022), ZebraLogic (Lin et al., 2025), and MATH500 (Hendrycks et al., 2021), we follow the standard single-answer setting and report **avg@1**.

**Compared Baselines.** We compare NPR with a broad set of strong baselines, covering open sequential reasoners, frontier parallel reasoners, and sequential training variants:

- **Open Sequential Reasoners:** We include Qwen2.5-32B-Instruct (Qwen et al., 2024), Qwen3-4B (Yang et al., 2025a) without thinking mode, and Qwen3-4B-Instruct-2507.
- **Frontier Parallel Reasoners:** We compare against Multiverse (Yang et al., 2025b) models, including Multiverse-32B and our reproduced Multiverse-4B based on Qwen3-4B-Instruct-2507.
- **Sequential Variants:** We further evaluate SR-BETA and SR, both of which are trained purely under the sequential reasoning paradigm.

### 4.3. Overall Reasoning Performance

The main experimental results are summarized in Table 2. Across all benchmarks, NPR demonstrates substantial gains over strong baselines (Qwen3-4B-Instruct-2507 and Qwen3-4B without thinking mode) and consistently outperforms both Multiverse-32B and Multiverse-4B.

**Training Data Advantage.** A major source of improvement comes from replacing the Multiverse training corpus (s1.1-8k for MV-4B) with our self-distilled dataset (orz-8k for NPR-BETA). Although the two pipelines differ slightly in implementation details, both adopt parallel-style SFT, making the comparison informative. The benefit of this data substitution is clear and consistent: performance improves from 46.7 to 50.8 on AIME24 (**+4.1**), from 60.2 to 76.1 on ZebraLogic (**+15.9**), from 75.0 to 85.9 on AMC23 (**+10.9**), and from 81.6 to 91.6 on MATH500 (**+10.0**). Overall, the average score increases from 50.1 to 59.0, yielding an absolute gain of **+8.9**.

*Table 2.* Performance of sequential and parallel reasoners on reasoning benchmarks. A25, A24, H25, OB, MvM, ZL, AMC23, and M500 denote AIME25, AIME24, HMMT25, OlympiadBench, Minerva-Math, ZebraLogic, AMC23, and MATH500, respectively. **S→P** indicates that MultiVerse transitions from sequential SFT to parallel SFT during training. Q2.5-32B-Inst., Q3-4B-Inst., and Q3-4B correspond to Qwen2.5-32B-Instruct, Qwen3-4B-Instruct-2507, and the Non-Thinking mode of Qwen3-4B. **MV** refers to the Multiverse models, and **SR** denotes Sequential Reasoner. "†" denotes the original results from the source work, and "-" indicates not available.

| Model | Data | Train | Base | A25 | A24 | H25 | OB | MvM | ZL | AMC23 | M500 | AVG |
|---|---|---|---|---|---|---|---|---|---|---|---|---|
| Q2.5-32B-Inst. | - | - | - | $10.4^\dagger$ | $15.8^\dagger$ | 3.8 | 46.4 | 40.8 | 43.6 | 62.8 | $80.4^\dagger$ | 38.0 |
| MV-32B | s1.1-8k | S→P SFT | Q2.5-32B-Inst. | $45.8^\dagger$ | $53.8^\dagger$ | 20.8 | 48.0 | 40.0 | 47.1 | 72.5 | $91.8^\dagger$ | 52.5 |
| Q3-4B-Inst. | - | - | - | $47.4^\dagger$ | 60.0 | $31.0^\dagger$ | 64.0 | 41.2 | $80.2^\dagger$ | 92.2 | 93.4 | 63.7 |
| MV-4B | s1.1-8k | S→P SFT | Q3-4B-Inst. | 42.9 | 46.7 | 20.8 | 38.8 | 34.9 | 60.2 | 75.0 | 81.6 | 50.1 |
| **NPR-BETA** | orz-8k | Parallel SFT | Q3-4B-Inst. | 42.9 | 50.8 | 23.3 | 60.1 | 41.2 | 76.1 | 85.9 | 91.6 | 59.0 |
| SR-BETA | | Sequential SFT | | 37.1 | 52.1 | 22.5 | 56.3 | 41.5 | 72.8 | 91.6 | 92.0 | 58.2 |
| SR | orz-8k | Sequential RL | NPR-BETA | 49.2 | 57.1 | 26.3 | 62.2 | 38.2 | 78.9 | 90.9 | 92.8 | 62.0 |
| **NPR** | | Parallel RL | | **50.4** | **63.3** | 30.8 | 63.7 | **43.0** | **81.7** | **93.1** | **93.6** | **65.0** |
| Q3-4B | - | - | - | $19.1^\dagger$ | $25.0^\dagger$ | $12.1^\dagger$ | 48.6 | 28.5 | $35.2^\dagger$ | 65.6 | 84.8 | 39.9 |
| NPR-BETA | orz-8k | Parallel SFT | Q3-4B | 43.8 | 52.5 | 29.2 | 57.8 | 45.9 | 70.0 | 85.3 | 86.8 | 58.9 |
| **NPR** | | Parallel RL | NPR-BETA | **53.8** | **62.5** | **32.9** | **61.9** | **47.1** | **75.8** | **89.7** | **91.8** | **64.4** |

*Takeaway:* These results indicate that our self-distillation corpus produces more accurate and diverse candidate solutions, whereas the Multiverse dataset, which was constructed from sequential reasoning traces, provides limited coverage of genuinely parallel reasoning patterns.

**Parallel SFT Advantage.** Switching from a sequential SFT procedure (*e.g.*, SR-BETA) to our parallel SFT approach (NPR-BETA) leads to consistent improvements across a variety of reasoning benchmarks. Sequential SFT imposes strong step-dependency priors, which limit flexible task decomposition. In contrast, our parallel SFT exposes the model to structurally parallel trajectories during training, enabling more independent subproblem exploration. Concretely, AIME25 improves from 37.1 to 42.9 (**+5.8**), OlympiadBench from 56.3 to 60.1 (**+3.8**), HMMT25 from 22.5 to 23.3 (**+0.8**), and ZebraLogic from 72.8 to 76.1 (**+3.3**). Overall performance increases from 58.2 to 59.0 (**+0.8**), with only minor regressions on a few benchmarks.

*Takeaway:* These findings demonstrate that parallel-format supervision encourages more adaptable and structurally diverse reasoning behaviors, alleviating the restrictive bias inherent in sequential SFT and improving robustness in downstream parallel generation.

**Parallel RL Advantage.** Building on NPR-BETA, our parallel RL algorithm brings further improvements and consistently outperforms sequential RL (NPR *vs.* SR). The gains are broad and systematic: AIME24 improves from 57.1 to 63.3 (**+6.2**), HMMT25 from 26.3 to 30.8 (**+4.5**), and Minerva-Math from 38.2 to 43.0 (**+4.8**). Other benchmarks also exhibit steady improvements, including AIME25 (**+1.2**), OlympiadBench (**+1.5**), ZebraLogic (**+2.8**), AMC23 (**+2.2**), and MATH500 (**+0.8**). Overall, the average score increases from 62.0 to 65.0 (**+3.0**).

*Takeaway:* The consistent improvements confirm that parallel RL more effectively amplifies high-reward reasoning modes learned during parallel SFT. Our PAPO and stable NPR Engine jointly enable reliable structural exploration and stronger performance across benchmarks.

## 5. Analyses and Discussion

### 5.1. Inference Acceleration While Improving Effectiveness

We evaluate token throughput and acceleration relative to Multiverse and autoregressive baselines. As reported in Table 3, our method achieves the best efficiency across all five benchmarks, consistently outperforming Multiverse ($1.3\times$ to $2.4\times$) and the autoregressive baselines, which demonstrates robust generalization. Importantly, speedup scales with task difficulty: we observe larger gains on harder problems (AIME25: $4.6\times$; HMMT25: $4.1\times$) than on easier ones (AMC23: $2.9\times$), indicating that our approach becomes increasingly advantageous when deeper exploration of solution paths is required. Combined with the effectiveness results in §4.3, these findings support the hypothesis that our method both improves accuracy and is especially effective when multiple solution strategies can be explored in parallel.

### 5.2. Parallel Reasoning Trigger Analysis

We quantify a model's tendency to produce simultaneous, non-sequential reasoning using the parallel reasoning trigger rate:

$$\text{parallel\_rate} = \frac{N_{\text{parallel}}}{N_{\text{total}}} \times 100\% \qquad (8)$$

where $N_{\text{parallel}}$ denotes the number of solutions exhibiting parallel reasoning and $N_{\text{total}}$ the total number of evaluated

*Table 3.* Evaluation results of tokens per second (TPS) and speedup ratio on selected benchmarks. The speedup ratio (denoted as **Speedup**) is calculated by comparing with sequential reasoning baseline.

| Method | AIME25 | | AIME24 | | HMMT25 | | AMC23 | | ZebraLogic | |
|---|---|---|---|---|---|---|---|---|---|---|
| | TPS | Speedup | TPS | Speedup | TPS | Speedup | TPS | Speedup | TPS | Speedup |
| SR | 646.8 | 1.0× | 667.5 | 1.0× | 683.8 | 1.0× | 685.5 | 1.0× | 649.7 | 1.0× |
| MULTIVERSE | 1579.0 | 2.4× | 1096.5 | 1.6× | 1465.1 | 2.1× | 1139.9 | 1.7× | 853.9 | 1.3× |
| NPR-Inst. | **2979.8** | **4.6×** | **2768.5** | **4.1×** | **2784.1** | **4.1×** | **1986.3** | **2.9×** | **2245.5** | **3.5×** |

*Table 4.* Comparison of parallel reasoning trigger rates between NPR and Multiverse across datasets.

| Model | AIME25 | AIME24 | HMMT25 | Olympiad | Minerva | ZebraLogic | AMC23 | MATH500 |
|---|---|---|---|---|---|---|---|---|
| MV-32B | 65.0 | 62.9 | 63.3 | 69.5 | 66.9 | 45.8 | 70.0 | 76.0 |
| NPR-Inst. | **100.0** | **100.0** | **100.0** | **100.0** | **100.0** | **100.0** | **100.0** | **100.0** |

test cases. Table 4 reports the parallel rate for the Multiverse baseline (MV-32B) and our NPR model (NPR-Inst.) across eight benchmark sets (AIME25, AIME24, HMMT25, OlympiadBench, Minerva, ZebraLogic, AMC23, and MATH500).

MV-32B exhibits substantial variation in parallel rate across datasets, indicating that its adoption of parallel reasoning is highly dataset-dependent. In particular, its parallel rate on logic-intensive tasks such as ZebraLogic is markedly lower than on several math competition benchmarks. This suggests that the Multiverse training paradigm, which gradually shifts the model from sequential to parallel behavior, may lead to incomplete or inconsistent internalization of parallel reasoning strategies and remain sensitive to domain-specific task characteristics.

By contrast, our NPR model attains a uniform **100.0%** parallel rate across all eight datasets. This consistency implies that the end-to-end NPR training pipeline more reliably institutionalizes parallel reasoning as the model's default problem-solving mode, independent of dataset domain or complexity. Practically, this means NPR not only triggers parallel reasoning more often, but does so robustly across heterogeneous evaluation sets.

### 5.3. Evolution Dynamics Towards NPR

As shown in Figure 4, the evolution toward native parallel reasoning (NPR) is gradual and structured. Naively enforcing the parallel generation format at the outset severely degrades performance (for example, Qwen3-4B-Instruct-2507 falls on AIME25 from 47.5% to 17.5%). To address this, we adopt a three-stage pipeline. *Stage 1* applies format-following reinforcement learning to stabilize format compliance and correctness, producing reliable trajectories that serve only as training data for the next stage[5]. *Stage*

---

[5]Stages 1 and 2 are trained from the same initialization; Stage 1 supplies data to Stage 2

*2* performs a parallel warmup via supervised fine-tuning, teaching independent branching and correct special-token usage, this structured learning causes a small, transient performance dip. Finally, *Stage 3* uses Native Parallel RL to recover and enhance reasoning quality, yielding final results that surpass the autoregressive baseline. Together, the results show that NPR is not merely a consequence of format supervision but emerges from progressively aligning format, parallel structure, and adaptive policy learning.

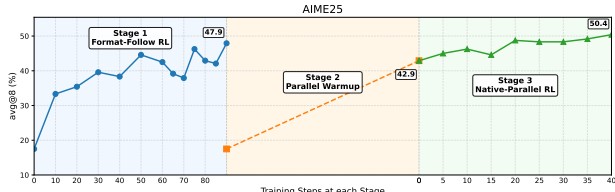

*Figure 4.* Evolving dynamics of evaluation on AIME 2025.

## 6. Conclusion

This work presents a simple and scalable framework for building a Native Parallel Reasoner that learns adaptive decomposition, diverse parallel planning, and reliable aggregation without relying on external teacher models. By combining self-distilled parallel SFT with agentic parallel RL, our approach produces genuinely parallel reasoning policies rather than simulated or scripted ones. Experiments on eight reasoning benchmarks show consistent improvements over Multiverse datasets, autoregressive training, and direct RL. Our analysis further demonstrates meaningful inference acceleration, stronger test-time scalability, and the absence of pseudo-parallel behavior. Case studies illustrate how the model adapts its parallelism to problem difficulty, enabling structured exploration and robust verification. These results indicate that native parallel reasoning is a promising direction for more general and scalable intelligence.

## Impact Statement

This paper presents work whose goal is to advance the field of Machine Learning. There are many potential societal consequences of our work, none which we feel must be specifically highlighted here.

## Acknowledgments

We thank the reviewers for their insightful suggestions to improve the manuscript. This work presented herein is supported by the National Natural Science Foundation of China (62376031).

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

# A. Algorithms

---

**Algorithm 1** Parallel Attention Mask

---

**Input:** sequence: $\mathcal{I} := \{t_1, \ldots, t_L\}$;
   Tag tokens: $\{\tau_{\text{parallel}}^{\pm}, \tau_{\text{step}}^{\pm}, \tau_{\text{plan}}^{\pm}\}$.
**Output:** Attention mask: $\mathbf{M} \in \mathbb{R}^{L \times L}$.

 1: **procedure** CONSTRUCT NPR ATTN MASK
 2:  $\mathbf{M} \leftarrow \text{tril}(\mathbf{1}_{L \times L})$                       $\triangleright$ Causal mask
 3:  $\mathcal{S} \leftarrow \emptyset$                          $\triangleright$ Init structure stack
 4:  **for** $i = 1 \ldots L$ **do**
 5:    **if** $t_i \in \{\tau_{\text{parallel}}^{+}, \tau_{\text{step}}^{+}, \tau_{\text{plan}}^{+}\}$ **then**
 6:      $\mathcal{S}.\text{push}(\{\text{type}(t_i), i\})$
 7:    **else if** $t_i \in \{\tau_{\text{step}}^{-}, \tau_{\text{plan}}^{-}\}$ **then**
 8:      $b \leftarrow \mathcal{S}.\text{pop}()$
 9:      Save span $(b.\text{start}, i)$ in parent block
10:    **else if** $t_i = \tau_{\text{parallel}}^{-}$ **then**
11:      $b \leftarrow \mathcal{S}.\text{pop}()$
12:      $\{\mathcal{P}_j = [s_j, e_j]\}_{j=1}^{n} \leftarrow b.\text{steps}$
13:      **for** $(j, k) \in [1, n]^2$ where $j \neq k$ **do**
14:        $\mathcal{I}_j \leftarrow \{s_j, \ldots, e_j - 1\}$
15:        $\mathcal{I}_k \leftarrow \{s_k, \ldots, e_k - 1\}$
16:        $\mathbf{M}[\mathcal{I}_j, \mathcal{I}_k] \leftarrow 0$               $\triangleright$ Isolate steps
17:        $\mathbf{M}[\mathcal{I}_k, \mathcal{I}_j] \leftarrow 0$
18:  $\mathbf{M} \leftarrow \begin{cases} 0 & \text{if } \mathbf{M}[i, j] = 1 \\ -\infty & \text{if } \mathbf{M}[i, j] = 0 \end{cases}$
19:  **return** $\mathbf{M}$

---

**Algorithm 2** Parallel Positional Encoding

---

**Input:** Token sequence $\mathcal{I} := \{t_1, \ldots, t_L\}$; Tag tokens $\{\tau_{\text{parallel}}^{\pm}, \tau_{\text{step}}^{\pm}, \tau_{\text{guideline}}^{\pm}\}$
**Output:** Position IDs: $\mathbf{P} \in \mathbb{R}^{L}$

 1: **procedure** CONSTRUCT NPR POSITION IDs
 2:  $\mathbf{P} \leftarrow [0, 1, \ldots, L-1]; \mathcal{S} \leftarrow \emptyset$        $\triangleright$ Init sequential positions & block stack
 3:  **for** $i = 1 \ldots L$ **do**
 4:    $b \leftarrow \mathcal{S}.\text{top}()$ if $\mathcal{S} \neq \emptyset$
 5:    **if** $t_i = \tau_{\text{guideline}}^{+}$ **then** $\mathcal{S}.\text{push}(\{p_{\text{end}} : -1, \ell_{\text{max}} : 0\})$    $\triangleright$ Open new <guideline> block
 6:    **else if** $t_i = \tau_{\text{guideline}}^{-}$ **then** $b.p_{\text{end}} \leftarrow \mathbf{P}[i]$       $\triangleright$ Mark <guideline> end position
 7:    **else if** $t_i = \tau_{\text{step}}^{+}$ and $b.p_{\text{end}} \geq 0$ **then** $\mathbf{P}[i:] \leftarrow \mathbf{P}[i:] - (\mathbf{P}[i] - b.p_{\text{end}} - 1)$   $\triangleright$ Reset to end
 8:    **else if** $t_i = \tau_{\text{step}}^{-}$ **then** $b.\ell_{\text{max}} \leftarrow \max(b.\ell_{\text{max}}, \mathbf{P}[i] - b.p_{\text{end}})$   $\triangleright$ Track length of max step
 9:    **else if** $t_i = \tau_{\text{parallel}}^{-}$ **then** $\mathbf{P}[i:] \leftarrow \mathbf{P}[i:] - (\mathbf{P}[i] - b.p_{\text{end}} - b.\ell_{\text{max}} - 1)$  $\triangleright$ Align to max
10:    $\mathcal{S}.\text{pop}()$                      $\triangleright$ Close <guideline> block
11:  **return** $\mathbf{P}$

---

# B. Test-time Scalability

We evaluate NPR's test-time scalability using the avg@8 and best@8 scores reported in Table 5. The results show that NPR reliably increases oracle coverage at test time, with the largest and most consistent gains occurring when the base model is relatively weak. For the Non-thinking backbone, supervised fine-tuning raises best@8 on AIME25 from 36.7 to 70.0, and NPR further increases it to 76.7, a **6.7** point improvement over SFT. On HMMT25 for the same backbone, best@8 moves from 23.3 to 46.7 after SFT and then to 53.3 with NPR, a further **6.6** points. For the Instruct backbone, NPR raises AIME25 best@8 to 70.0 compared with 63.3 for SFT. Overall, NPR amplifies the coverage benefits introduced by SFT and converts modest increases in sample diversity into meaningful gains in best@8, although the magnitude of improvement depends on the task and the starting strength of the backbone.

*Table 5.* Performance shown for SFT and RL checkpoints on both the Instruct and Non-thinking Qwen3-4B backbones.

| | AIME25 | | AIME24 | | HMMT25 | | AMC23 | |
|---|---|---|---|---|---|---|---|---|
| | avg@8 | best@8 | avg@8 | best@8 | avg@8 | best@8 | avg@8 | best@8 |
| Qwen3-4B-Instruct-2507 | 47.4 | 63.3 | 60.0 | **86.7** | 31.0 | 46.7 | 92.2 | 96.7 |
| NPR-BETA-Inst. | 42.9 | 63.3 | 50.8 | 83.3 | 23.3 | 46.7 | 85.9 | 97.5 |
| NPR-Inst. | 50.4 | 70.0 | **63.3** | 80.0 | 30.8 | **53.3** | **93.1** | **100.0** |
| Qwen3-4B-Non-thinking | 19.1 | 36.7 | 25.0 | 40.0 | 12.1 | 23.3 | 65.6 | 93.3 |
| NPR-BETA-Non. | 43.8 | 70.0 | 52.5 | 83.3 | 29.2 | 46.7 | 85.3 | 97.5 |
| NPR-Non. | **53.8** | **76.7** | 62.5 | 80.0 | **32.9** | **53.3** | 89.7 | **100.0** |

## C. Qualitative Case Studies

Table 6 and Table 7 present two representative examples that highlight how our system performs parallel reasoning. Both solutions are generated entirely through the parallel inference pipeline and are answered correctly. A key property of the system is its adaptivity: more challenging problems automatically trigger broader exploration, while easier prompts elicit only a small number of focused branches. The model organizes its computation into parallel blocks, with the final block often functioning as a multi-angle verifier that cross-checks or refines candidate solutions.

Parallelism operates in two complementary ways. *First*, it can explore multiple independent solution strategies simultaneously. *Second*, it can decompose a single problem into smaller subproblems that are solved in parallel. For correct examples, at least one branch converges to a valid answer, and the verification block preserves it. Errors, when they occur, typically arise from an incorrect high-level plan that propagates through many branches or from branches that expand into unnecessarily long, error-prone reasoning chains. By exploring diverse trajectories and applying lightweight consistency checks, the parallel architecture provides substantially greater robustness than relying on a single sequential reasoning path.

*Table 6.* Example of decomposing a problem into multiple subproblems for independent resolution, followed by reflective reasoning in the `<takeaway>` block.

---

**Case #1**

---

**Problem**: What is the domain of the function $f(x) = \frac{2-x}{\log(2-\log(x-2))}$, where $\log$ is the base 10 logarithm function? Express your answer in interval notation.

**NPR Generation**:

**<guideline>**

**<plan>**1: Determine the domain by identifying all values of $x$ for which the denominator is defined and non-zero, and the logarithmic expressions are valid.**</plan>**

**<plan>**2: Break down the inner logarithm $\log(x-2)$ and the outer logarithm $\log(2-\log(x-2))$ to find the required domain constraints.**</plan>**

**<plan>**3: Ensure that the argument of every logarithm is positive and that the entire denominator is not zero.**</plan>**

**<guideline>**

**<step>**1: $\cdots$, Therefore, $x = 12$ must be excluded from the domain.**</step>**

**<step>**2: $\cdots$, So the domain is $(2, 102) \setminus \{12\}$, or $(2, 12) \cup (12, 102)$.**</step>**

**<step>**3: $\cdots$, Thus, the condition $x \neq 12$ does not introduce any new restriction because $x = 12$ is already excluded by $x < 10$. So no additional point needs to be removed from the domain.**</step>**

**<takeaway>**$\cdots$, which gives $x - 2 < 100$, so $x < 102$, **but wait — this is not tight. Wait — re-examine**: We said $\log(x-2) < 2 \Rightarrow x - 2 < 100 \Rightarrow x < 102$, **which is correct. But is that the only constraint?** Actually, $x - 2 > 0 \rightarrow x > 2$, and $x < 102$, so domain is $(2, 102)$, $\cdots$, **But wait — what about the inner logarithm?** We need $x > 2$, which is already covered. $\cdots$, Therefore, the domain is $(2, 12) \cup (12, 102)$.**</takeaway>**

**<guideline>**

**<plan>**1: Verify the domain by testing values within the proposed intervals.**</plan>**

**<plan>**2: Confirm that all conditions (logarithm domains, non-zero denominator) are fully captured.**</plan>**

**<guideline>**

**<step>**1: $\cdots$, All tests confirm that the domain is $(2, 12) \cup (12, 102)$.**</step>**

**<step>**2: $\cdots$, No other points cause issues. The logarithmic functions are defined in the intervals, and no other values make the denominator zero. Thus, the domain is fully described.**</step>**

**<takeaway>**All constraints have been verified through both algebraic reasoning and numerical testing. $\cdots$**</takeaway>**

The domain of the function $f(x) = \frac{2-x}{\log(2-\log(x-2))}$ is $\boxed{(2, 12) \cup (12, 102)}$.

---

*Table 7.* Examples of exploring different methods to independently solve problems and engaging in reflective reasoning within the `<step>`.

---

**Case #2**

---

**Problem**: Altitudes $\overline{AD}$ and $\overline{BE}$ of $\triangle ABC$ intersect at $H$. If $\angle BAC = 54°$ and $\angle ABC = 52°$, then what is $\angle AHB$?

**NPR Generation**:

`<guideline>`

`<plan>`1: Use the property that in any triangle, the angle between two altitudes intersecting at the orthocenter can be found using the relationship involving the angles of the triangle.`</plan>`

`<plan>`2: Apply the fact that in triangle $ABC$, with altitudes $AD$ and $BE$ intersecting at $H$, the quadrilateral $ADHE$ is cyclic, and use angle chasing in this cyclic quadrilateral.`</plan>`

`<plan>`3: Use the known result that $\angle AHB = 180° - \angle C$, where $H$ is the orthocenter of triangle $ABC$.`</plan>`

`<guideline>`

`<step>`1: $\cdots$, Therefore, $\angle AHB = 180° - 74° = 106°$. `</step>`

`<step>`2: $\cdots$, but note that $\angle AED = 180° - \angle ABE = 180° - 90° + \angle ABC = 90° + 52° = 142°$? **Wait—better approach:** in the cyclic quadrilateral formed by the feet and orthocenter, the angle at $H$ can be found via inscribed angle relationships. **Alternatively, consider** triangle $ABC$, $\cdots$, then $\angle AHB = 180° - 74° = 106°$. `</step>`

`<step>`3: $\cdots$, Therefore, $\angle AHB = 180° - \angle ACB = 180° - 74° = 106°$.`</step>`

`<takeaway>`All three approaches lead to the same conclusion: $\cdots$, **The reasoning is consistent across different strategies**: either using known orthocenter angle properties, cyclic quadrilateral angle chasing, or direct application of the formula. **All confirm the same result.**`</takeaway>`

`<guideline>`

`<plan>`1: Verify the result by direct angle chasing in quadrilateral $ADHE$. `</plan>`

`<plan>`2: Cross-check with triangle angle sum. `</plan>`

`<guideline>`

`<step>`1: $\cdots$, **Wait — actually,** $\angle ABD$ is not necessarily that. **Wait:** $\angle ABD$ is part of $\angle ABC = 52°$. Actually, since $AD$ is an altitude from $A$, $\cdots$, **Wait —** in triangle $ABC$, $AD \perp BC$, so $D$ lies on $BC$, $BE \perp AC$, so $E$ lies on $AC$. Then quadrilateral $BDHE$ has right angles at $D$ and $E$, so $BDHE$ is cyclic. Therefore, $\angle BHD = 180° - \angle BEC$? **Wait. Actually,** in cyclic quadrilateral $BDHE$, $\cdots$, and no contradiction arises from the other approaches, the conclusion stands.`</step>`

`<step>`2: $\cdots$, This confirms the earlier conclusion.`</step>`

`<takeaway>`**All lines of reasoning**—whether through known orthocenter properties, cyclic quadrilateral angle chasing, or direct triangle angle sum—**lead to the same result**: $\angle AHB = 106°$. **The result is consistent, reliable, and internally verified.**`</takeaway>`

The measure of $\angle AHB$ is $\boxed{106°}$.

---

