# OpenReview forum: "Native Parallel Reasoner: Reasoning in Parallelism via Self-Distilled Reinforcement Learning"
_ICML.cc/2026/Conference — ICML 2026 regular_

### Official Review · Reviewer_V2FA · 2026-02-15

**Soundness:** 3
**Presentation:** 3
**Significance:** 3
**Originality:** 3
**Overall Recommendation:** 4
**Confidence:** 4

**Summary:**

This paper proposes the Natural Parallel Reasoner (NPR), a novel fine-tuning method that enables language models to perform parallel reasoning. First, NPR trains language models via reinforcement learning to sequentially output "map-process-reduce" patterns, which generates a dataset for subsequent training. Next, utilizing this dataset, NPR trains the models to generate outputs in parallel through special attention mask patterns. Finally, NPR applies reinforcement learning to further boost generalization performance. Empirically, NPR demonstrates superior performance compared to the prior parallel reasoning method, Multiverse, on various mathematical reasoning benchmarks.

**Compliance With Llm Reviewing Policy:**

Affirmed.

**Final Justification:**

The authors successfully addressed my primary concern regarding fair comparisons by aligning the training datasets and providing wall-clock inference times. Consequently, I have raised my score to a Weak Accept. However, the lack of a direct comparison with sequential reasoning approaches limits the paper's scope strictly to parallel reasoning. Because of this narrow scope, I cannot recommend a full Accept, but the work remains a solid contribution within its specific domain.

**Key Questions For Authors:**

I am willing to raise my score if these questions, along with the weaknesses mentioned above, are adequately addressed.
- Table 4 reports a parallel reasoning trigger rate, but a higher trigger rate is not necessarily better. Even if parallel reasoning is triggered, the branches could be near-duplicates, resulting in redundant computation rather than meaningful exploration. Did the authors measure the diversity of parallel branches? If not, it would be helpful to discuss how one could quantify branch diversity and distinguish "meaningful" parallelism from redundant branching. No additional experiments are required.
- I am curious whether the Stage 2 self-distilled SFT step is strictly necessary. Is there a reason the training cannot transition directly from Stage 1 to Stage 3? For example, one could define a curriculum that gradually interpolates between sequential and parallel attention masks and run Stage 3 RL under this curriculum. I would appreciate the authors' perspective on why Stage 2 is important (or why a direct transition from Stage 1 to Stage 3 might fail). No additional experiments are required.

**Limitations:**

Yes.

**Strengths And Weaknesses:**

I think the paper is overall well written, well motivated, and technically strong.
- The proposed three-stage training pipeline (RL → SFT → RL), which does not rely on proprietary LLMs, is a novel and meaningful contribution.
- The paper clearly diagnoses the failure modes of applying RL to parallel reasoning and proposes simple, elegant remedies.
- Beyond the algorithmic ideas, the paper also provides system-level engineering optimizations that make parallel reasoning stable and scalable in practice.

However, the paper also has several weaknesses that should be addressed to strengthen the empirical claims and overall presentation.
- The authors should make a stronger effort to ensure a fair, apples-to-apples comparison between NPR and Multiverse. At minimum, it would be helpful to train the NPR using the same s1.1-8k dataset.
- The paper does not appear to report the average number of generated tokens. This is important for a fair efficiency comparison. Even if parallel reasoning achieves higher tokens per second than sequential reasoning, this advantage can be offset if it generates more tokens overall, resulting in little or no improvement in end-to-end wall-clock time.
- I believe the evaluation would be stronger if the authors also compared against methods that reduce the effective context window in sequential reasoning (e.g., The Markovian Thinker [1]).
- In the avg@k evaluation, k = 8 seems too small. Please consider increasing it to k = 32 or reporting the standard deviation.
- The section titles are not consistent in capitalization. Some headings use title case while others use inconsistent or mixed capitalization which hurts overall polish. The same capitalization inconsistency also appears in the paragraph headings.
- There are some LaTex issues. Please ensure the necessary packages are properly included for \mathbb{1} in Equation 3 and 4. In Equation 2 and 5, please use \ldots instead of \cdots.

[1] Aghajohari et al., The Markovian Thinker: Architecture-Agnostic Linear Scaling of Reasoning, ICLR 2026.

---

> ### Author Rebuttal · Authors · 2026-03-30
>
> We are grateful to the reviewer for his/her insightful review. The reviewer's suggestions have provided valuable guidance for improving the paper.  We provided detailed responses to each point below.
>
> ---
>
> **Q**: Ensure a fairer comparison between NPR and Multiverse, ideally by training NPR on the same s1.1-8k dataset.
>
> **A**: We would like to clarify that the two data sources are not directly comparable in quality: s1.1 is a carefully cleaned dataset, whereas the 8k ORZ subset in our experiments is randomly sampled mainly to keep the training scale aligned.
>
> This difference is also consistent with prior results on the same base model, Qwen2.5-32B-Instruct: training on s1.1-8k achieves 56.7 on AIME24, while training on the full 57k ORZ data reaches 48.1, which suggests that s1.1 is a substantially stronger training source.
>
> To enable a fairer apples-to-apples comparison under matched training data, we have added results for Multiverse (MV-4B) trained on ORZ-8k. As shown in the table below, under the same training set, NPR still shows a significant advantage over Multiverse on both AIME24 and AIME25. These results further support that NPR is more effective than Multiverse under the same training data setting.
>
> **Table 1**: MV-4B vs. NPR-4B.
>
> |        | A25  | A24  |
> |--------|------|------|
> | MV-4B  | 40.4 | 51.7 |
> | NPR-4B | 53.8 | 62.5 |
>
> ---
>
> **Q**: The paper does not report average generated length, which matters for efficiency comparison because higher TPS may not reduce end-to-end latency if more tokens are generated overall.
>
> **A**: We agree that TPS measures throughput rather than end-to-end latency. In our experiments, all methods use the same maximum generation length (30,000) and have similar output lengths, so higher TPS also translates into substantial wall-clock speedup. We will clarify this in the revision and report wall-clock inference time below (8 NVIDIA A100, TP=2, batch size=1):
>
> **Table 2**: Wall-clock inference time of MV-4B and NPR-4B.
>
> |         | A25     | A24     | H25     | AMC23   | ZL      |
> |---------|---------|---------|---------|---------|---------|
> | MV-4B   | 71.5 min| 66.2 min| 52.0 min| 13.1 min| 58.1 min|
> | NPR-4B  | 15.2 min| 17.4 min| 12.8 min| 6.4 min | 23.9 min|
> | Speedup | 4.7×    | 3.8×    | 4.1×    | 2.0×    | 2.4×    |
>
> ---
>
> **Q**: The evaluation could be strengthened by comparing with sequential methods that reduce the effective context window, such as The Markovian Thinker.
>
> **A**: We agree that The Markovian Thinker is a very excellent contemporaneous work. That paper studies a complementary direction: making sequential reasoning scale linearly by resetting context and carrying forward a compact textual state, whereas our focus is native parallel reasoning under strict branch isolation, so the two approaches are largely orthogonal and could potentially be combined in future work. We will cite this paper in the revision and discuss its relation to our method more explicitly.
>
> ---
>
> **Q**: In the avg@k evaluation, k = 8 seems too small. Consider increasing it to k = 32 or reporting the standard deviation.
>
> **A**: We use avg@8 mainly to follow the Multiverse protocol for fair comparison, and this setting is also common in recent reasoning work. We therefore add the standard deviation results below:
>
> **Table 3**: avg@8 with standard deviation.
>
> |        | A25         | A24         | H25         | AMC23       |
> |--------|-------------|-------------|-------------|-------------|
> | MV-32B | 45.8 ± 3.83 | 53.8 ± 3.59 | 20.8 ± 1.93 | 72.5 ± 3.08 |
> | NPR-4B | 50.4 ± 3.21 | 63.3 ± 3.17 | 30.8 ± 1.64 | 93.1 ± 3.25 |
>
> ---
>
> **Q**: The section and paragraph headings use inconsistent capitalization, and there are some LaTeX issues.
>
> **A**: We will standardize the capitalization and fix the LaTeX issues in the revision.
>
> ---
>
> **Q**: Discuss how to distinguish meaningful parallelism from redundant parallelism.
>
> **A**: Based on our observations and the cases in the paper, we did not observe branch duplication in practice, though incorrect branches do occur. We do not explicitly measure branch diversity or redundancy in the current submission, but agree this is a valuable direction.
>
> One possible approach is to measure semantic and structural differences across branches, and assess whether they provide complementary reasoning for the final reduce step rather than redundant content.
>
> ---
>
> **Q**: Is Stage 2 necessary, or could training transition directly from Stage 1 to Stage 3?
>
> **A**: In our experience, Stage 2 is important for two reasons. (1) RL alone does not reliably learn the required special-token format, so the model cannot consistently identify valid parallel trigger points. (2) RL provides only outcome-level supervision, and a direct transition to Stage 3 often produces structurally noisy rollouts, making parallel attention masks and position ids difficult to define. Stage 2 therefore bridges Stage 1 formatting and stable parallel trajectories.

---

> > ### Author Rebuttal · Reviewer_V2FA · 2026-04-02
> >
> > The authors clearly address my questions, providing detailed explanation on wall-clock inference time analysis, branch duplication, and importance of stage 2 in the proposed algorithm. Now, this paper is definitely above the acceptance threshold.

---

> > > ### Author Response · Authors · 2026-04-05
> > >
> > > We appreciate the reviewers' positive feedback and the higher score.

---

### Official Review · Reviewer_GnkN · 2026-02-19

**Soundness:** 3
**Presentation:** 3
**Significance:** 3
**Originality:** 3
**Overall Recommendation:** 5
**Confidence:** 5

**Summary:**

The paper proposes Native Parallel Reasoner (NPR), a teacher-free training framework that makes autoregressive LLMs perform truly parallel reasoning via a branched decoding graph rather than sequential “simulated” parallelism. It uses a Map–Process–Reduce tagged reasoning schema and a three-stage curriculum: RL to induce the format (NPR-ZERO), rejection sampling + parallel SFT with parallel-aware attention/position encoding (NPR-BETA), and a final native-parallel RL stage (NPR). The authors introduce Parallel-Aware Policy Optimization (PAPO) to optimize branching and aggregation decisions directly within parallel rollouts. They also build an NPR Engine to make large-scale parallel decoding practical and stable. Across multiple reasoning benchmarks on a Qwen3-4B backbone, NPR reports higher accuracy together with substantial inference speedups (up to ~4.6×).

**Compliance With Llm Reviewing Policy:**

Affirmed.

**Final Justification:**

The author's response resolved my concern.

**Key Questions For Authors:**

1. How do you verify that the model is executing native parallel reasoning rather than producing a parallel-looking format while still relying on hidden sequential dependencies?
2. What are the main failure modes you observe in practice (e.g., branch inconsistency, reduce-step hallucination, schema gaming), and how frequently do they occur across tasks?

**Limitations:**

yes

**Strengths And Weaknesses:**

### Strengths

1. The paper proposes a clear end-to-end training pipeline that directly targets true parallel reasoning rather than sequential imitation. The three stages are logically connected and each stage addresses a concrete limitation of the previous one. Overall, the method feels internally consistent and implementable.
2. The work pays attention to practical training stability for parallel rollouts, not just high-level ideas. It identifies real failure modes in parallel decoding and provides fixes that make the approach workable at scale. This engineering grounding increases confidence that the method can run reliably.
3. The paper includes helpful implementation details and behavior analyses beyond accuracy numbers. It reports both performance and efficiency indicators, making the evaluation more complete. Code/model release also improves accessibility for readers.

### Weaknesses

1. The presentation sometimes mixes core method description with implementation-specific details, which can blur the main narrative. Readers may struggle to distinguish the conceptual contribution from engineering constraints. This reduces clarity about what is fundamental versus incidental.
2. How do you conceptualize and control the trade-off between exploration (more branches) and coherence (global consistency) during decoding?

---

> ### Author Rebuttal · Authors · 2026-03-30
>
> We sincerely thank the reviewer for the thorough and thoughtful evaluation. We are encouraged that the reviewer recognizes NPR as a promising direction for native parallel reasoning, and we greatly appreciate the careful attention to both the technical design and the broader implications of our work. Below, we provide point-by-point responses to the concerns and suggestions.
>
> ---
>
> **Q**: The presentation sometimes mixes core method description with implementation-specific details, which can blur the main narrative. Readers may struggle to distinguish the conceptual contribution from engineering constraints. This reduces clarity about what is fundamental versus incidental.
>
> **A**: Thank you for this helpful suggestion. In the revision, we will explicitly distinguish the core algorithmic ideas from the NPR Engine, which is introduced as an enabling backend for stable large-scale parallel RL.
>
> ---
>
> **Q**: How do you conceptualize and control the trade-off between exploration (more branches) and coherence (global consistency) during decoding?
>
> **A**: Thank you for this important question. In our framework, this trade-off is controlled implicitly by the **adaptive branching policy itself**: the model is encouraged to explore diverse candidate branches, but all branches must eventually be reconciled through the shared reduce stage, and PAPO optimizes this full execution graph end-to-end rather than rewarding branching in isolation.
>
> Empirically, we also observe that the model adapts its degree of parallelism across tasks instead of simply increasing branch count.
>
> ---
>
> **Q**: How do you verify that the model is executing native parallel reasoning rather than producing a parallel-looking format while still relying on hidden sequential dependencies?
>
> **A**: Thank you for raising this important point. Our key distinction from pseudo-parallel reasoning is that, in our setting, **different branches at each parallel step receive different inputs** and are **enforced to be strictly mutually invisible**, so they cannot rely on hidden sequential information exchange while only appearing parallel on the surface.
>
> Moreover, during evaluation, all cases are executed under this strict format without any manual intervention, and the model performs native parallel reasoning directly under the same constraints rather than reverting to an underlying sequential process.
>
> ---
>
> **Q**: What are the main failure modes you observe in practice (e.g., branch inconsistency, reduce-step hallucination, schema gaming), and how frequently do they occur across tasks?
>
> **A**: Thank you for this practical question. The main failure modes we observed are primarily discussed at the infrastructure level, including incorrect parallel token accounting leading to overlong generation, runtime instability from adaptive parallel inference logic, and memory-management issues such as radix-cache-related leaks; these motivate the NPR Engine redesign, while at the modeling level our pseudo-parallel analysis was intended to guard specifically against schema gaming and AR fallback.
>
> ---
>
> We hope our answers have resolved your concerns.

---

> > ### Author Rebuttal · Reviewer_GnkN · 2026-04-01
> >
> > The author's response resolved my concern.

---

> > > ### Author Response · Authors · 2026-04-05
> > >
> > > We appreciate the reviewers' positive feedback and the higher score.

---

### Official Review · Reviewer_N1ii · 2026-03-13

**Soundness:** 4
**Presentation:** 3
**Significance:** 2
**Originality:** 2
**Overall Recommendation:** 4
**Confidence:** 5

**Summary:**

This paper presents a highly solid and well-executed study on native parallel reasoning. While the core idea itself may not feel especially novel or elegant, the work stands out for its strong soundness, careful problem framing, and thorough engineering realization. In particular, the authors do not merely propose a parallel reasoning concept, but systematically address the full pipeline: from training curriculum and parallel execution design to reinforcement learning stabilization and system-level implementatio, which resulting in strong empirical performance. Overall, this is a paper where effort and execution quality outweigh formulation novelty, but the final package is nevertheless quite convincing and effective.

**Compliance With Llm Reviewing Policy:**

Affirmed.

**Final Justification:**

The author's response resolved my concern, so i will keep my positive score, and recommend its acceptance.

**Key Questions For Authors:**

No major questions.

**Limitations:**

yes

**Strengths And Weaknesses:**

## Strengths

1. A major strength of the paper is its strong end-to-end engineering completeness. Rather than stopping at a high-level idea, the work carefully addresses the practical issues required to make native parallel reasoning function in practice, including curriculum design, parallel warm-up, RL adaptation, and serving/rollout support. This makes the paper feel much more mature and credible than many works that only present a conceptual prototype.

2. The paper demonstrates a strong understanding of prior system and training bottlenecks. The authors clearly identify why naive parallelization, teacher-dependent supervision, or directly applying standard RL pipelines are insufficient, and they propose a coherent solution stack to address these issues. This contributes significantly to the soundness of the work.

3. The work is empirically strong and technically thorough. The combination of modeling, RL, and system optimization leads to a convincing overall package, and the reported gains in both effectiveness and efficiency suggest that the proposed design is not merely conceptually interesting, but practically useful.


## Weaknesses

The main weakness is that the initial guideline-based formulation feels somewhat heavy-handed and not fully native. In particular, the use of explicit guidelines introduces a strong structural prior, which makes the resulting reasoning process feel less organic than the paper’s broader framing of “native” parallel reasoning might suggest. While this design is likely useful for stabilization and training, it is also the least elegant part of the method.

---

> ### Author Rebuttal · Authors · 2026-03-30
>
> We sincerely thank the reviewer for the thoughtful comments and for recognizing NPR as a promising direction. Our response is as follows:
>
> ---
>
> **Q**: The main weakness is that the initial guideline-based formulation feels somewhat heavy-handed and not fully native. In particular, the use of explicit guidelines introduces a strong structural prior, which makes the resulting reasoning process feel less organic than the paper’s broader framing of “native” parallel reasoning might suggest. While this design is likely useful for stabilization and training, it is also the least elegant part of the method.
>
> **A**: We suspect there may be a slight mismatch in how “native parallel” is being interpreted. In our view, native parallel reasoning does not mean removing all explicit structure; rather, it means that the model itself decides whether to branch, how many branches to create, what plan each branch should follow, and how to summarize them into a final conclusion. In other words, **the entire Map–Process–Reduce procedure is driven by the model rather than externally scripted.**
>
> From this perspective, **the format used in our method serves only as a minimal execution interface**, rather than constituting the essence of native parallelism. The special tokens are introduced to indicate when parallel execution should begin and when reduction should occur. For example, once the model generates a branching trigger, the system can instantiate mutually invisible parallel branches; once it generates the conclusion trigger, the system can enter the aggregation stage. Without such markers, there is no reliable way to realize actual parallel execution.
>
> Therefore, these tokens are introduced to **enable** native parallel reasoning, not to define it, nor merely to stabilize training. What is native in NPR is that the model autonomously performs decomposition, branch-level reasoning, and final aggregation under strict parallel constraints, rather than following a hand-crafted branching policy.
>
> We will clarify this point more explicitly in the revision, and we hope our response has addressed your concern.

---

> > ### Author Rebuttal · Reviewer_N1ii · 2026-03-31
> >
> > The author's response resolved my concern.

---

> > > ### Author Response · Authors · 2026-04-05
> > >
> > > Thank you very much for your positive evaluation. As the other two reviewers have raised their scores following our rebuttal, we sincerely hope you might consider raised your score as well if our responses have addressed your concerns.

---

### Decision · Program_Chairs · 2026-04-30

**Decision:**

Accept (regular)

**Comment:**

The reviewers are unanimously positive regarding the "Native Parallel Reasoner" (NPR). The consensus is that this is an exceptionally well-executed study on parallel reasoning that moves beyond conceptual prototyping to provide a full, stable engineering realization.

The paper is technically solid and provides a convincing solution to the challenge of making parallel reasoning "native" to autoregressive models. While some initial concerns were raised regarding the "guideline-heavy" nature of the method and the fairness of comparisons to sequential baselines, the authors' rebuttal successfully addressed these by providing wall-clock latency data and aligned dataset evaluations. The work represents a significant contribution to the field of efficient reasoning, balancing theoretical motivation with strong system-level execution.